# Undirected Probabilistic Model for Tensor Decomposition

**Zerui Tao**[1,2]    **Toshihisa Tanaka**[1,2]    **Qibin Zhao**[2,1]*

zerui.tao@riken.jp    tanakat@cc.tuat.ac.jp    qibin.zhao@riken.jp

[1]Tokyo University of Agriculture and Technology    [2]RIKEN AIP

## Abstract

Tensor decompositions (TDs) serve as a powerful tool for analyzing multiway data. Traditional TDs incorporate prior knowledge about the data into the model, such as a directed generative process from latent factors to observations. In practice, selecting proper structural or distributional assumptions beforehand is crucial for obtaining a promising TD representation. However, since such prior knowledge is typically unavailable in real-world applications, choosing an appropriate TD model can be challenging. This paper aims to address this issue by introducing a flexible TD framework that discards the structural and distributional assumptions, in order to learn as much information from the data. Specifically, we construct a TD model that captures the joint probability of the data and latent tensor factors through a deep energy-based model (EBM). Neural networks are then employed to parameterize the joint energy function of tensor factors and tensor entries. The flexibility of EBM and neural networks enables the learning of underlying structures and distributions. In addition, by designing the energy function, our model unifies the learning process of different types of tensors, such as static tensors and dynamic tensors with time stamps. The resulting model presents a doubly intractable nature due to the presence of latent tensor factors and the unnormalized probability function. To efficiently train the model, we derive a variational upper bound of the conditional noise-contrastive estimation objective that learns the unnormalized joint probability by distinguishing data from conditional noises. We show advantages of our model on both synthetic and several real-world datasets.

## 1 Introduction

Tensor decompositions (TDs) serve as powerful tools for analyzing high-order and high-dimensional data, aiming to capture the inter-dependencies among different modes by utilizing multiple latent factors. TDs have demonstrated remarkable success in various machine learning tasks, including data imputation [53, 9], factor analysis [4], time-series forecasting [27], model compression [28, 41], generative models [10, 20] among others.

Existing TDs typically incorporate predefined directed graphical models into the generative process. These models specify the priors of latent factors and the conditional probabilities of observations, following specific contraction rules associated with the latent factors. Traditional contraction rules predominantly employ multi-linear products, like CP [15], Tucker [42], tensor train [29] and other variants [19, 6]. However, selecting an appropriate contraction rule for specific datasets is often challenging in real-world applications. Recent research, known as tensor network structure search [TNSS, 22, 23], has demonstrated that selecting an appropriate TN contraction rule significantly enhances the factorization performance. Another promising approach involves learning non-linear mappings from the data, utilizing techniques like nonparametric models [5, 45, 53] and deep neural networks

---

*Corresponding author

37th Conference on Neural Information Processing Systems (NeurIPS 2023).

[25, 9]. Empirical results demonstrate that non-linear TDs exhibit superior performance compared to traditional multi-linear TDs in various applications, attributed to their enhanced expressive power.

Despite the success of non-linear TDs in reducing structural assumptions, they often rely on simplistic distributional assumptions. Typically, a specific directed graphical model is adopted to model the generative process from latent factors to tensor entries, represented as $p(x) = \int p(x \mid z)p(z)\,\mathrm{d}z$, where $z$ denotes tensor factors and $x$ represents observations. Additionally, the distributions are usually selected from exponential families for tractability, such as Gaussian and Bernoulli distributions. For instance, a Gaussian prior can be assigned to latent factors, and observed entries can be modeled using Gaussian distribution [32, 49] or Gaussian process [45, 53]. However, these prior assumptions regarding the probabilistic model can introduce model bias and reduce the effectiveness of TD models. In real-world applications, the latent factors might originate from unknown distributions, and the observations can exhibit complex multi-modal generative processes. Without knowing the underlying generative process, these simplistic assumptions can lead to inaccurate estimations.

To address these issues, this paper proposes to construct an undirected graphical model of TD. More specifically, a TD model that captures the joint probability of the data and latent tensor factors is constructed through a deep energy-based model (EBM), represented as $p(x, z) \approx \exp(-f(x, z))$. Neural networks (NNs) are then employed to parameterize the joint energy function $f(x, z)$. The flexibility of EBM and NNs facilitates the learning of underlying structures and distributions. Furthermore, our model unifies the learning process in the presence of side information, such as dynamic tensors with time stamps, by designing the energy function. The resulting model presents a doubly intractable nature due to the presence of latent tensor factors and the unnormalized probability density function (pdf). For efficient model training, we derive a variant of conditional noise-contrastive estimation [3] algorithm that learns the unnormalized joint probability by distinguishing data from conditional noises. The proposed model offers several advantages: (1) it features a flexible structure that can adapt to different distributions; (2) the undirected nature allows us to learn more general correlations than traditional directed TDs; (3) it can handle diverse tasks and encode auxiliary information by adjusting the energy function.

Experiments are conducted on synthetic and real-world datasets to showcase the advantages of our model. Through simulation studies, we demonstrate the capability of our model to handle data generated from diverse distributions, in contrast to traditional Gaussian-based models that yield unfaithful and biased estimates. Subsequently, experiments are performed on multiple real-world datasets to evaluate sparse and continuous-time tensor completion. Our model outperforms various baselines across multiple metrics and settings, highlighting the generality of the proposed model.

## 2 Backgrounds

**Notations**   We adopt similar notations with [19]. Throughout the paper, we use lowercase letters, bold lowercase letters, bold capital letters and calligraphic bold capital letters to represent scalars, vectors, matrices and tensors, *e.g.*, $x, \boldsymbol{x}, \boldsymbol{X}$ and $\boldsymbol{\mathcal{X}}$. Tensors refer to multi-way arrays which generalize matrices. For a $D$-order tensor $\boldsymbol{\mathcal{X}} \in \mathbb{R}^{I_1 \times \cdots \times I_D}$, we denote its $(i_1, \ldots, i_D)$-th entry as $x_{\mathbf{i}}$.

### 2.1 Tensor decomposition

Given a $D$-order tensor $\boldsymbol{\mathcal{X}} \in \mathbb{R}^{I_1 \times \cdots \times I_D}$, tensor decomposition (TD) aims to factorize $\boldsymbol{\mathcal{X}}$ into $D$ smaller latent factors $\boldsymbol{Z}^{d=1,\ldots,D} \in \mathbb{R}^{I_d \times R_d}$ by using some predefined tensor contraction rules. The classical Tucker decomposition [42] assumes $\boldsymbol{\mathcal{X}} = \boldsymbol{\mathcal{W}} \times_1 \boldsymbol{Z}^1 \times_2 \cdots \times_D \boldsymbol{Z}^D$, where $\boldsymbol{\mathcal{W}} \in \mathbb{R}^{R_1 \times \cdots \times R_D}$ is the coefficient and $\times_d$ denotes the matrix-tensor contraction [19]. Equivalently, each entry can be written as $x_{\mathbf{i}} = \sum_{r_1=1}^{R_1} \cdots \sum_{r_D=1}^{R_D} w_{r_1 \ldots r_D} z_{i_1 r_1}^1 \cdots z_{i_D r_D}^D$, where the tuple $(R_1, \ldots, R_D)$ is the Tucker rank of tensor $\boldsymbol{\mathcal{X}}$. The latent factors $\boldsymbol{Z}^d$ can capture information of each tensor mode and $\boldsymbol{\mathcal{W}}$ represents the weight of each factors. CP decomposition [15] is a restricted form of Tucker by assuming $\boldsymbol{\mathcal{W}}$ is super-diagonal, *i.e.*, $x_{\mathbf{i}} = \sum_{r=1}^{R} w_r z_{i_1 r}^1 \cdots z_{i_D r}^D$, where we simplify $w_r = w_{r \ldots r}$. In this paper, we focus on probabilistic version of TDs, which serves as generalizations of traditional ones. The standard approach is to formulate TDs as a directed graphical model, $p(\boldsymbol{\mathcal{X}}) = \int p(\boldsymbol{\mathcal{X}} \mid \boldsymbol{Z})p(\boldsymbol{Z})\,\mathrm{d}\boldsymbol{Z}$, where $\boldsymbol{Z}$ denotes $\{\boldsymbol{Z}^1, \ldots, \boldsymbol{Z}^D\}$ for simplicity. For continuous data, the $p(\boldsymbol{\mathcal{X}} \mid \boldsymbol{Z})$ is usually assumed to be Gaussian and TDs are used to parameterize the mean of corresponding Gaussian distribution [32, 49, 50].

Despite the elegant form of these multi-linear contraction rules, they have limited flexibility that can be mitigated by extending TDs to their non-linear counterparts. We can think of TD as a function that maps the multiway latent factors to tensor entries. One extension is to add Gaussian process (GP) priors on the function to obtain a nonparametric model, which resembles a GP latent variable model. In particular, [53] proposed to stack the latent factors as $\boldsymbol{m_i} = [\boldsymbol{z}_{i_1}^1, \ldots, \boldsymbol{z}_{i_D}^D] \in \mathbb{R}^{DR}$ and then assign a GP prior on the functional mapping. In specific, for continuous data, it assumes $x_{\mathbf{i}} \sim \mathcal{N}(\mu_{\mathbf{i}}, \sigma^2)$, where the mean function is a GP $\mu_{\mathbf{i}} = f(\boldsymbol{m_i}) \sim \mathcal{GP}(0, k(\boldsymbol{m_i}, \cdot))$ associated with kernel $k(\cdot, \cdot)$. Since GP has large computational complexity and designing the kernel requires ad-hoc expert domain knowledge, [25] proposed to parameterize the function using neural networks (NNs), $x_{\mathbf{i}} \sim \mathcal{N}(\mu_{\mathbf{i}}, \sigma^2)$ where $\mu_{\mathbf{i}} = f_{\mathrm{NN}}(\boldsymbol{m_i})$. However, NNs easily overfit due to the high-dimensional and sparse nature of tensor data. To address this issue, [9] proposed to use Bayesian NN with spike-and-slab prior for sparse weights. All these models are based on directed graphical model, that assumes there exists a direct mapping from the latent factors to tensor entries and use simplistic distributions.

## 2.2 Energy-based model

Energy-based model [EBM, 21] is a class of undirected probabilistic models, which uses an energy function to characterize the data distribution. Given observed data $x$, the basic idea of EBM is to approximate the data distribution by a Boltzmann distribution, $p_{\mathrm{data}}(x) \approx \frac{\exp(-f(x;\theta))}{Z(\theta)}$, where $Z(\theta) = \int \exp(-f(x;\theta)) \, \mathrm{d}x$ is the normalization constant (a.k.a., the partition function) and $f(x;\theta)$ is the energy function. One classical example of EBM is the restricted Boltzmann machine (RBM) where the energy function has bi-linear form for tractability. In deep EBMs, the energy function is typically parameterized by deep neural networks. The main difficulty for training EBMs is to deal with the intractable normalization constant [36]. There are several ways to train EBMs, including contrastive divergence [14], score matching [16], noise-contrastive estimation [NCE, 11] and so on. In this paper, we focus on NCE, due to its efficiency and ability of tackling different data types.

We denote the unnormalized pdf as $\phi(x;\theta) = \exp(-f(x;\theta))$. NCE consider the normalization constant $Z(\theta)$ as a trainable parameter. However, maximum likelihood estimation (MLE) does not work for this case since $Z(\theta)$ can be arbitrarily small and the log-likelihood goes to infinity. Instead, the NCE can be obtained by maximizing the following objective,

$$\mathcal{L}_{\mathrm{NCE}}(\theta) = \mathbb{E}_x \log h(x;\theta) + \nu \mathbb{E}_y \log(1 - h(y;\theta)),$$

where $x$ denotes observed data and $y$ denotes noises generated from some known distribution $p_n$, $\nu$ is the ratio between noise and sample sizes, *i.e.*, $\nu = \#y / \#x$. And $h(\cdot)$ can be regarded as a classifier that distinguish data from noises, defined as follows, $h(u;\theta) = \frac{\phi(u;\theta)}{\phi(u;\theta) + \nu p_n(u)}$. It has been shown that NCE is consistent with MLE [11].

Although the noise distribution is essential for training efficiency, the selection of noises is currently limited to heuristics. A common intuitive is that the noise should be similar with the data. Indeed, [3] proposed to use conditional noises $y \sim p_c(y \mid x)$ and minimizing the following loss function,

$$\mathcal{L}_{\mathrm{CNCE}}(\theta) = 2\mathbb{E}_{xy} \log[1 + \exp(-G(x, y))], \tag{1}$$

where $G(u_1, u_2; \theta) = \log \frac{\phi(u_1;\theta) p_c(u_2 | u_1)}{\phi(u_2;\theta) p_c(u_1 | u_2)}$ with $y$ drawn from $p_c(y \mid x)$.

## 3 Proposed model

### 3.1 Energy-based tensor decomposition

Even though many non-linear tensor decompositions (TD) have been proposed to enhance flexibility, existing methods typically adopt simplistic distributions such as Gaussian. This can be problematic for complex real-world data. To address the issue, we propose energy-based tensor decomposition (EnergyTD), by integrating EBMs in TD framework. Given an order-$D$ tensor $\boldsymbol{\mathcal{X}}$ of shape $I_1 \times \cdots \times I_D$, we aim to factorize it into $D$ smaller latent factors $\boldsymbol{Z}^d \in \mathbb{R}^{I_d \times R}, \forall d = 1, \ldots, D$. We denote the latent factor associated with the $(i_1, \ldots, i_D)$-th entry as $\boldsymbol{m_i} = [\boldsymbol{z}_{i_1}^1, \ldots, \boldsymbol{z}_{i_D}^D] \in \mathbb{R}^{DR}$, where $\boldsymbol{z}_{i_d}^d \in \mathbb{R}^R$ is the $i_d$-th row of $\boldsymbol{Z}^d$. Unlike traditional directed TDs trying to parameterize the conditional expectation $\mathbb{E}[x_{\mathbf{i}} \mid \boldsymbol{m_i}] = f(\boldsymbol{m_i})$, we model the joint distribution using an EBM,

$$p(x_{\mathbf{i}}, \boldsymbol{m_i}; \theta) = \frac{\exp(-f(x_{\mathbf{i}}, \boldsymbol{m_i}; \theta))}{Z(\theta)}, \tag{2}$$

where $f(\cdot, \cdot; \theta)$ is the energy function and $Z(\theta) = \int \exp(-f(x_\mathbf{i}, \boldsymbol{m}_\mathbf{i}; \theta)) \, \mathrm{d}x_\mathbf{i} \, \mathrm{d}\boldsymbol{m}_\mathbf{i}$ is the partition function to make it a valid pdf. We further assume the joint probability of all entries are independent, *i.e.*, $p(\boldsymbol{\mathcal{X}}, \boldsymbol{m}) = \prod_{\mathbf{i} \in \Omega} p(x_\mathbf{i}, \boldsymbol{z}_{i_1}^1, \ldots, \boldsymbol{z}_{i_D}^D)$, where $\Omega$ denotes the set of observed entries. This is a standard setting in TDs and the dependence of tensor entries can be captured by sharing latent factors.

The expressive nature of the energy function enables us to easily handle diverse data types. For example, we can deal with discrete data by plugging one-hot codings into Eq. (2) to represent categorical probabilities. Additionally, the flexibility of NNs allows us to model tensors with side information, where each tensor entry incorporates additional features [34]. Specifically, in this paper, we focus on a particular case of dynamic tensors with continuous time stamps [48]. In this case, we consider an observed tensor as a time series $\boldsymbol{\mathcal{X}}_t$, where the time stamp $t$ is continuous, and each entry $\mathbf{i}$ has its own specific time stamp $t_\mathbf{i}$. To model the tensor time series, we assume that each entry follows the same distribution and construct the time-dependent energy function, $p(x_\mathbf{i}, \boldsymbol{m}_\mathbf{i}; \theta, t_\mathbf{i}) \propto \exp(-f(x_\mathbf{i}, \boldsymbol{m}_\mathbf{i}, t_\mathbf{i}; \theta))$, where the time stamp $t_\mathbf{i}$ is considered as an auxiliary feature. The flexibility of NNs allows this function to learn general patterns across continuous time stamps. Experimental results demonstrate that this simple treatment can achieve good performances.

**Network architecture** The network architecture plays a crucial role in learning accurate probabilistic manifolds. Specifically, we define the energy function as $f(x_\mathbf{i}, \boldsymbol{m}_\mathbf{i}) = g_1(g_2(g_3(x_\mathbf{i}), g_4(\boldsymbol{m}_\mathbf{i})))$, where $g_3$ and $g_4$ are MLP layers that encode information from $x_\mathbf{i}$ and $\boldsymbol{m}_\mathbf{i}$, respectively. $g_2$ is a summation or concatenation layer that induce coupling between tensor values and latent factors, and $g_1$ is the output layer. Although we currently utilize only MLPs, it is worth noting that convolutional architectures, as demonstrated by [39, 25], can also be employed, which is a topic for future research. To handle dynamic tensors, we incorporate an extra sinusoidal positional encoding layer [37] denoted as $g_5(t)$ to capture temporal information. This embedding utilizes random Fourier features as proposed by [31]. Consequently, the energy function can be expressed as $f(x_\mathbf{i}, \boldsymbol{m}_\mathbf{i}, t_\mathbf{i}) = g_1(g_2(g_3(x_\mathbf{i}), g_4(\boldsymbol{m}_\mathbf{i}), g_5(t_\mathbf{i})))$. This architecture is commonly employed to capture temporal information and has been demonstrated to effectively learn high-frequency information when combined with MLPs [37].

**Posterior sampling** A significant application of TDs is to estimate the posterior of missing entries. Unlike traditional TDs, direct predictions cannot be obtained even after learning the latent factors due to the utilization of an undirected probabilistic model. Instead, we need to seek for sampling methods of $p(x_\mathbf{i} \mid \boldsymbol{m}_\mathbf{i})$. One choice is score-based samplers by utilizing the score function $\nabla_{x_\mathbf{i}} \log p(x_\mathbf{i} \mid \boldsymbol{m}_\mathbf{i}) = \nabla_{x_\mathbf{i}} \log \frac{p(x_\mathbf{i}, \boldsymbol{m}_\mathbf{i})}{p(\boldsymbol{m}_\mathbf{i})} = -\nabla_{x_\mathbf{i}} f(x_\mathbf{i}, \boldsymbol{m}_\mathbf{i})$, such as Langevin dynamics [44]. Score-based samplers are not suitable for handling discrete data. However, in our case, we model the one-dimensional pdf for each entry, enabling us to directly sample the discrete data. Consequently, for continuous data, the use of grid search is a viable approach to obtain maximum a posteriori (MAP) estimations.

## 3.2 Learning objective

Despite the flexibility of the proposed model in Eq. (2), obtaining maximum likelihood estimation (MLE) becomes doubly intractable, as both the partition function $Z(\theta)$ and the marginal distribution $p(x_\mathbf{i})$ are intractable. Therefore, the CNCE loss Eq. (1) cannot be directly applied. In this section, we extend the variational approach [33] to construct a upper bound that addresses the challenge posed by intractable marginal distributions.

Denote the unnormalized pdf as $\phi(x_\mathbf{i}, \boldsymbol{m}_\mathbf{i}; \theta) = \exp(-f(x_\mathbf{i}, \boldsymbol{m}_\mathbf{i}; \theta))$ and the unnormalized marginal pdf as $\phi(x_\mathbf{i}; \theta) = \int \phi(x_\mathbf{i}, \boldsymbol{m}_\mathbf{i}; \theta) \, \mathrm{d}\boldsymbol{m}_\mathbf{i}$. For clarity, we omit the index $\mathbf{i}$ in the subsequent context of this subsection. We follow the idea of CNCE [3] to distinguish data $x$ from conditional noises $y \sim p_c(y \mid x)$. Firstly, Eq. (1) can be rewritten as

$$\mathcal{L}_{\mathrm{CNCE}}(\theta) = 2\mathbb{E}_{xy} \log[1 + 1/r(x, y; \theta)], \tag{3}$$

where

$$r(x, y; \theta) = \frac{\phi(x; \theta) p_c(y \mid x)}{\phi(y; \theta) p_c(x \mid y)}. \tag{4}$$

However, for our problem, the unnormalized marginal probability $\phi(x; \theta)$ is unknown. An additional variational distribution $q(\boldsymbol{m}; \varphi)$ is used to approximate the true posterior $p(\boldsymbol{m} \mid \boldsymbol{\mathcal{X}}; \theta)$. Note that in TDs where the data size is static, there is no need for amortized inference, which is different

from previous ones like [2]. Equipped with the variational distribution, the unnormalized marginal distribution can be computed using importance sampling,

$$\phi(x;\theta) = \int \frac{\phi(x,\boldsymbol{m};\theta)q(\boldsymbol{m};\varphi)}{q(\boldsymbol{m};\varphi)} \, \mathrm{d}\boldsymbol{m} = \mathbb{E}_{q(\boldsymbol{m};\varphi)}\left[\frac{\phi(x,\boldsymbol{m};\theta)}{q(\boldsymbol{m};\varphi)}\right]. \tag{5}$$

Plugging Eq. (5) into Eq. (4), we have

$$r(x,y;\theta) = \frac{\mathbb{E}_{q(\boldsymbol{m};\varphi)}[\phi(x,\boldsymbol{m};\theta)/q(\boldsymbol{m};\varphi)]p_c(y \mid x)}{\phi(y;\theta)p_c(x \mid y)}. \tag{6}$$

Since Eq. (3) is a convex function w.r.t. $r(x,y;\theta)$, plugging Eq. (6) into Eq. (3) and applying the Jensen's inequality, we have the upper bound,

$$\mathcal{L}_{\mathrm{CNCE}}(\theta) = 2\mathbb{E}_{xy}\log[1 + 1/r(x,y;\theta)]$$
$$\leq 2\mathbb{E}_{xy}\mathbb{E}_{q(\boldsymbol{m};\varphi)}\log\left[1 + \frac{\phi(y;\theta)p_c(x \mid y)q(\boldsymbol{m};\varphi)}{\phi(x,\boldsymbol{m};\theta)p_c(y \mid x)}\right] \triangleq \mathcal{L}_{\mathrm{VCNCE}(\theta,\varphi)}. \tag{7}$$

Following [33], we have the theorem about the tightness of the bound.

**Theorem 1** *The difference between the VCNCE loss Eq. (7) and CNCE loss Eq. (1) is the expectation of the $f$-divergence,*

$$\mathcal{L}_{\mathrm{VCNCE}}(\theta,\varphi) - \mathcal{L}_{\mathrm{CNCE}}(\theta) = \mathbb{E}_{xy}[\mathbb{D}_{f_{xy}}(p(\boldsymbol{m} \mid x;\theta)\|q(\boldsymbol{m};\varphi))],$$

*where $f_{xy}(u) = \log(\frac{\kappa_{xy}+u^{-1}}{\kappa_{xy}+1})$ with $\kappa_{xy} = \frac{\phi(x;\theta)p_c(y|x)}{\phi(y;\theta)p_c(x|y)}$.*

The proof can be found in appendix. Based on the theorem, we have the following corollaries to justify the optimization process.

**Corollary 1** *When $q(\boldsymbol{m};\varphi)$ equals to the true posterior, the CVNCE bound is tight, i.e.,*

$$\mathcal{L}_{\mathrm{VCNCE}} = \mathcal{L}_{\mathrm{CNCE}} \iff q(\boldsymbol{m};\varphi) = p(\boldsymbol{m} \mid x;\theta).$$

**Corollary 2** *The following two optimization problems are equivalent,*

$$\min_{\theta} \mathcal{L}_{\mathrm{CNCE}}(\theta) = \min_{\theta} \min_{q(\boldsymbol{m};\varphi)} \mathcal{L}_{\mathrm{VCNCE}}(\theta,\varphi).$$

In practice, we need to seek for the sampled version of Eq. (7). Supposing we have $N$ observed samples $\{x_{\mathbf{i}}\}_{\mathbf{i}=1}^{N}$, $\nu$ noises $\{y_{\mathbf{i},j}\}_{j=1}^{\nu}$ for each sample $x_{\mathbf{i}}$ and using importance sampling for $\phi(y;\theta)$, the sampled objective function is,

$$\mathcal{L}_{\mathrm{VCNCE}}(\theta,\varphi) = \frac{2}{\nu N}\sum_{\mathbf{i}=1}^{N}\sum_{j=1}^{\nu}\mathbb{E}_{q(\boldsymbol{m}_{\mathbf{i}};\varphi)}\log\left[1 + \frac{\mathbb{E}_{q(\boldsymbol{m}_{\mathbf{i}};\varphi)}\left[\frac{\phi(y_{\mathbf{i},j},\boldsymbol{m}_{\mathbf{i}};\theta)}{q(\boldsymbol{m}_{\mathbf{i}};\varphi)}\right]p_c(x_{\mathbf{i}} \mid y_{\mathbf{i},j})q(\boldsymbol{m}_{\mathbf{i}};\varphi)}{\phi(x_{\mathbf{i}},\boldsymbol{m}_{\mathbf{i}};\theta)p_c(y_{\mathbf{i},j} \mid x_{\mathbf{i}})}\right].$$

Specifically, we formulate $q(\boldsymbol{m};\varphi)$ as a diagonal Gaussian and use reparameterization trick [18] to compute the expectation. When dealing with continuous data, we typically select conditional Gaussian noises, represented as $p_c(y \mid x) = \mathcal{N}(y \mid x,\sigma^2)$. This choice entails only one hyperparameter $\sigma$ that needs to be tuned. Another benefit is the symmetry of the conditional distribution for Gaussian noise, expressed as $p_c(y \mid x) = p_c(x \mid y)$. Hence, the objective function can be further reduced. For binary or categorical data, such symmetric noises can also be derived [3].

The time complexity of the proposed objective is $\mathcal{O}(\nu B(DRH + LH^2))$, where $B$ is the batch size, $\nu$ is the number of conditional noises, $H$ is the number of hidden units per layer, $L$ is the number of layers and $D$ is the tensor order. The time complexity of our model is $\nu$ times greater than traditional TDs, since we need to compute forward passes for $\nu$ particles. However, as we only use small networks, the computational speed is still very fast (See Appendix C.3 for an illustration).

# 4 Related work

Traditional tensor decompositions (TDs) are based on multi-linear contraction rules, such as CP [15], Tucker [42], tensor networks [29, 51] and their variations [19, 6]. In this paper, we mainly focus on probabilistic TDs, which extend traditional methods by providing uncertainty estimates about both observations and latent factors [32, 49, 50, 26, 38]. These models build directed mapping from latent factors to tensor entries using multi-linear contraction rules, resulting in limited flexibility when dealing with complex datasets. An alternative approach involves replacing the multi-linear relations with non-linear ones. [5, 45, 52] introduced the use of tensor-variate Gaussian processes (GPs) for achieving nonparametric factorization. [53] further expanded on this concept by incorporating a GP prior on the function that maps latent factors to tensor entries, resulting in a nonparametric TD for sparse tensors. GP-based TDs are further extended using hierarchical priors [40], stochastic processes [43, 8]. Despite the success of GP-based TDs, nonparametric approaches can encounter computational challenges and may still lack sufficient flexibility. Recently, neural networks (NNs) are also applied to TDs. [25] suggested the utilization of convolutional NNs to map latent factors to tensor entries. Besides, [7] built a hierarchical version of the Tucker model and introduced non-linear mappings within each hierarchy of latent factors. To mitigate overfitting, [39] suggested the adoption of deep kernels in GP-based TD rather than using NNs directly. On the other hand, [9] proposed to use Bayesian NN with spike-and-slab prior to prevent from overfitting and obtain probabilistic estimates. More recently, [24] adopt neural ODEs to capture dynamic tensor trajectories. Other works regarding more flexible exponential families [13] or mixture of Gaussians [12] employ linear structures. While all these methods using directed mapping from latent factors to tensor entries, our model is fundamentally different from them, in that we construct much more flexible undirected probabilistic model of TD that can deal with diverse distributions and structures.

Another related direction is the energy-based model (EBM). To address the intractable pdf of EBMs, various training methods have been proposed, including contrastive divergence [CD, 14], score matching [SM, 16], noise-contrastive estimation [NCE, 11]. CD requires large steps of Monte Carlo samples, which can be computationally expensive for high-dimensional tensors. SM cannot handle discrete data, and learning latent variables with SM requires complex bi-level optimization [2]. Therefore, we focus on NCE in this paper. Learning energy-based TD is even more challenging because it involves multiple coupled latent factors that cannot be analytically marginalized. [33] proposed VNCE to handle unnormalized models with latent factors. We enhance their algorithm by using conditional noises [3] to improve the learning efficiency. One fundamental distinction between our model and traditional EBMs is that, through TD construction, we only need to learn one-dimensional distributions for each scalar entry, instead of the original high-dimensional tensor. Hence, our model avoids performance degradation when learning high-dimensional data using NCE.

# 5 Experiments

We demonstrate the proposed energy-based TD (EnergyTD) on synthetic data and several real-world applications. All the experiments are conducted on a Linux workstation with Intel Xeon Silver 4316 CPU, 256GB RAM and NVIDIA RTX A5000 GPUs (24GB memory each). The code is implemented based on PyTorch 1.12.1 [30]. More experimental details can be found in the appendix. The code is available at https://github.com/taozerui/energy_td

## 5.1 Simulation study

**Tensors with non-Gaussian distributions**  Traditional TDs commonly assume that tensor entries follow a Gaussian distribution. However, in real-world applications, we often encounter highly complex distributions. In this experiment, we evaluate the capability of our model to learn distributions that deviate from the Gaussian assumption.

We consider a two-mode tensor of shape $I \times I$, where we set $I = 8$. Firstly, two latent factors of shape $I \times R$ are generated, where the rank $R$ is set to 5. Then, conditioned on the latent factors, we generated tensor observations from particular distributions. For each entry, we generate $N = 200$ samples. Three types of distributions are considered: (1) Beta distribution; (2) Mixture of Gaussians (MoG) and (3) Exponential distribution. For Beta distribution, we generate latent factors from uniform distribution $\boldsymbol{Z}^{i=1,2} \overset{\text{iid}}{\sim} Uni(0.0, 1.1)$. Then, we sample the observed tensor from Beta

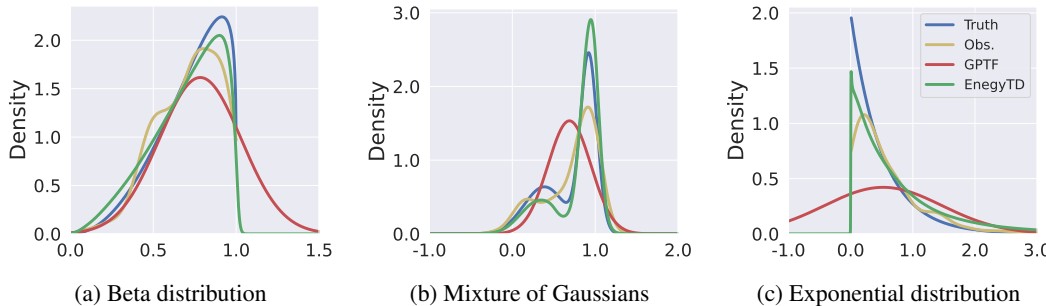

(a) Beta distribution  (b) Mixture of Gaussians  (c) Exponential distribution

Figure 1: Simulation results for different distributions. The blue line is the ground truth pdf. The yellow line is the kernel density estimation (KDE) plot of observed samples. The red line is the GPTF estimation, which is a Gaussian pdf. The green line is our method, computed by evaluating the unnormalized pdf on grids and calculating the partition function using Gaussian quadrature.

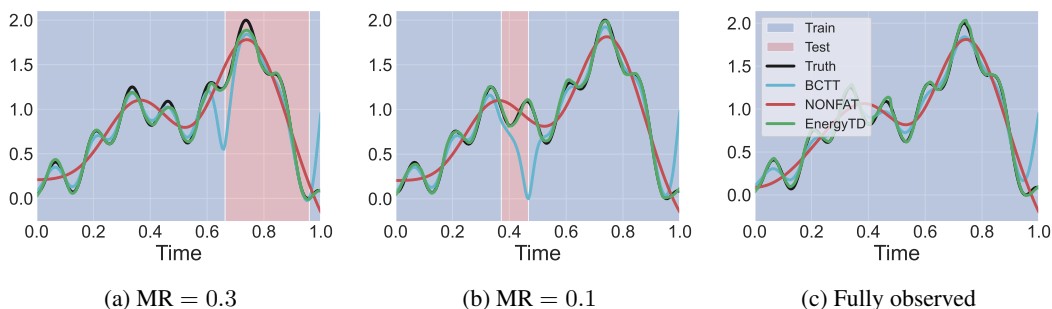

(a) MR = 0.3  (b) MR = 0.1  (c) Fully observed

Figure 2: Simulation results for continuous-time tensor decomposition. The blue regions are observed and the red regions are missing. The trajectories of ground truth, BCTT, NONFAT and our model are drawn in black, blue, red and green lines, respectively.

distribution $x_{ij} \stackrel{\text{iid}}{\sim} Beta((\boldsymbol{Z}^1\boldsymbol{Z}^{2,\top})_{ij}, 1.2)$. For MoG distribution, we draw latent factors from uniform distribution $\boldsymbol{Z}^{i=1,2} \stackrel{\text{iid}}{\sim} Uni(0.0, 1.0)$. The tensor entries are then drawn from MoG $x_{ij} \stackrel{\text{iid}}{\sim}$ $0.6 \cdot \mathcal{N}(\cos((\boldsymbol{Z}^1\boldsymbol{Z}^{2,\top})_{ij}), 0.1^2) + 0.4 \cdot \mathcal{N}(\sin((\boldsymbol{Z}^1\boldsymbol{Z}^{2,\top})_{ij}), 0.25^2)$. For Exponential distribution, we generate latent factors from uniform distribution $\boldsymbol{Z}^{i=1,2} \stackrel{\text{iid}}{\sim} Uni(0.0, 1.0)$. The tensor entries are then sampled from Exponential distribution $x_{ij} \stackrel{\text{iid}}{\sim} Exp((\boldsymbol{Z}^1\boldsymbol{Z}^{2,\top})_{ij})$. We compare with GP tensor factorization [GPFT, 53], which assumes the entries follow Gaussian distribution.

The results of probability density function (pdf) estimation are presented in Fig. 1. We display the learned pdf of one single tensor entry. This reveals that GPTF is limited to capturing only the 1st moments (mean values) and overlooks higher-order information of more complex distributions. Our model exhibits greater flexibility in handling non-Gaussian distributions.

**Continuous-time tensors**  We then consider a dynamic tensor, where each entry is a time series. We follow similar setting with [9] and use the same data size with the previous simulation, *i.e.*, a two-mode tensor of shape $8 \times 8$ with each entry being a time series of length 200. We firstly generate latent factors of shape $8 \times 2$, with each rows drawn from $\boldsymbol{z}_i^1 \sim \mathcal{N}([0, 2], 2 \cdot \boldsymbol{I})$ and $\boldsymbol{z}_i^2 \sim \mathcal{N}([1, 1], 2 \cdot \boldsymbol{I})$. Then we generate $N = 200$ observed entries from time $t \in [0, 1]$. The tensor entries are computed by $x_{\mathbf{i}}(t) = \sum_{r_1=1}^{2} \sum_{r_2=1}^{2} z_{i_1 r_1}^1 z_{i_2 r_2}^2 \omega_{r_1 r_2}(t)$, where $\omega_{11}(t) = \sin(2\pi t), \omega_{12}(t) = \cos(2\pi t), \omega_{21}(t) = \sin^2(2\pi t)$ and $\omega_{22}(t) = \cos(5\pi t) \sin^2(5\pi t)$. Finally, the data are normalized to range $[0, 2]$. The synthetic data consist of low-frequency trends and high-frequency fluctuations. Apart from fully observed case, we also test with missing rates (MR) 10% and 30%. In specific, for each entry, we randomly select a starting time and set the following consecutive 10% or 30% time stamps as missing.

We compare with two methods that are designed for dynamic tensors, the Bayesian continuous-time Tucker decomposition [BCTT, 8] and the nonparametric factor trajectory learning [NONFAT, 43].

Table 1: Sparse tensor completion

| | RMSE | | | | MAE | | | |
|---|---|---|---|---|---|---|---|---|
| *Alog* | Rank 3 | Rank 5 | Rank 8 | Rank 10 | Rank 3 | Rank 5 | Rank 8 | Rank 10 |
| CP-WOPT | $1.486 \pm 0.282$ | $1.386 \pm 0.043$ | $1.228 \pm 0.063$ | $1.355 \pm 0.079$ | $0.694 \pm 0.098$ | $0.664 \pm 0.018$ | $0.610 \pm 0.027$ | $0.658 \pm 0.026$ |
| GPTF | $0.911 \pm 0.008$ | $0.867 \pm 0.008$ | $0.878 \pm 0.009$ | $0.884 \pm 0.009$ | $0.511 \pm 0.005$ | $0.494 \pm 0.004$ | $0.530 \pm 0.004$ | $0.554 \pm 0.006$ |
| HGP-GPTF | $0.896 \pm 0.011$ | $0.867 \pm 0.009$ | $0.850 \pm 0.011$ | $0.844 \pm 0.006$ | $0.479 \pm 0.007$ | $0.473 \pm 0.003$ | $0.474 \pm 0.004$ | $0.480 \pm 0.004$ |
| POND | $0.885 \pm 0.010$ | $0.871 \pm 0.013$ | $0.858 \pm 0.009$ | $0.857 \pm 0.011$ | $0.463 \pm 0.004$ | $0.454 \pm 0.005$ | $0.444 \pm 0.005$ | $0.443 \pm 0.006$ |
| CoSTCo | $0.999 \pm 0.007$ | $0.936 \pm 0.017$ | $0.930 \pm 0.024$ | $0.909 \pm 0.014$ | $0.523 \pm 0.006$ | $0.481 \pm 0.007$ | $0.514 \pm 0.031$ | $0.481 \pm 0.008$ |
| EnergyTD | $\mathbf{0.864 \pm 0.011}$ | $\mathbf{0.835 \pm 0.011}$ | $\mathbf{0.840 \pm 0.013}$ | $\mathbf{0.833 \pm 0.016}$ | $\mathbf{0.450 \pm 0.006}$ | $\mathbf{0.433 \pm 0.006}$ | $\mathbf{0.424 \pm 0.005}$ | $\mathbf{0.409 \pm 0.004}$ |
| *ACC* | | | | | | | | |
| CP-WOPT | $0.533 \pm 0.039$ | $0.592 \pm 0.037$ | $0.603 \pm 0.028$ | $0.589 \pm 0.022$ | $0.138 \pm 0.004$ | $0.147 \pm 0.005$ | $0.148 \pm 0.003$ | $0.147 \pm 0.004$ |
| GPTF | $0.367 \pm 0.001$ | $0.357 \pm 0.001$ | $0.359 \pm 0.001$ | $0.368 \pm 0.001$ | $0.152 \pm 0.002$ | $0.150 \pm 0.001$ | $0.167 \pm 0.002$ | $0.182 \pm 0.001$ |
| HGP-GPTF | $0.355 \pm 0.001$ | $0.344 \pm 0.001$ | $0.341 \pm 0.001$ | $0.338 \pm 0.001$ | $0.125 \pm 0.003$ | $0.129 \pm 0.001$ | $0.139 \pm 0.000$ | $0.145 \pm 0.002$ |
| CoSTCo | $0.385 \pm 0.003$ | $0.376 \pm 0.018$ | $0.363 \pm 0.004$ | $0.348 \pm 0.002$ | $0.117 \pm 0.004$ | $0.137 \pm 0.020$ | $0.107 \pm 0.004$ | $\mathbf{0.101 \pm 0.004}$ |
| EnergyTD | $\mathbf{0.348 \pm 0.005}$ | $\mathbf{0.336 \pm 0.004}$ | $\mathbf{0.328 \pm 0.003}$ | $\mathbf{0.328 \pm 0.003}$ | $\mathbf{0.110 \pm 0.008}$ | $\mathbf{0.101 \pm 0.006}$ | $\mathbf{0.094 \pm 0.006}$ | $\mathbf{0.101 \pm 0.009}$ |

BCTT treats Tucker core tensors as functions of time and NONFAT treats all GPTF factors as time series. Unlike BCTT with GP prior on the time domain, NONFAT uses GP prior on the frequency domain through inverse Fourier transform of original time series.

Fig. 2 displays the completion results. The learned trajectory of a single tensor entry is plotted. Higher missing rates result in the inability of BCTT to capture accurate trajectories, particularly in missing regions. NONFAT achieves more stable predictions, yet it tends to favor over-smoothed trajectories while disregarding high-frequency fluctuations. This behavior may be attributed to its unique construction, which introduces a GP prior in the frequency domain. Our utilization of flexible neural networks allows us to adapt to complex situations encompassing both low-frequency and high-frequency information.

## 5.2 Tensor completion

We evaluate our model on two sparse tensor and two dynamic tensor completion applications. For real datasets, the energy function can be difficult to learn, when the pdf has a very sharp curve. Motivated by the idea of noise-perturbed score estimation [35], we add small i.i.d. Gaussian noises on the data during the training of EnergyTD as a form of smoothing technique. The results are reported on *clean* test data. For EnergyTD, we use MAP estimates as described in Section 3.1.

### 5.2.1 Sparse tensor completion

We test our model on two sparsely observed tensors: (1) *Alog*, a file access log dataset [52] of shape *200 users × 100 actions × 200 resources* with about 0.33% nonzero entries; (2) *ACC*, a three-way tensor generated from a code repository management system [52] of shape *3k users × 150 actions × 30k resources* with about 0.009% nonzero entries. We use the same dataset split as in [52] and report the 5-fold cross validation results.

**Competing methods**   We compare with five baselines: (1) CP-WOPT [1], CP decomposition with stochastic optimization; (2) GPTF [53], a GP-based tensor factorization using stochastic variational inference; (3) HGP-GPTF [40], a GPTF equipped with hierarchical Gamma process prior; (4) POND [39], a probabilistic non-linear TD using deep kernels with convolutional NNs (CNNs); (5) CoSTCo [25], a non-linear TD that uses CNNs to map latent factors to tensor entries. CP-WOPT is provided in Matlab Tensor Toolbox [1]. We implement GPTF based on PyTorch by ourselves and use official implementations for HGP-GPTF[2], POND[3] and CoSTCo[4].

**Experimental settings and results**   We set batch size 1000 and run 1000 epochs for *Alog*, 100 epochs for *ACC*. For our model, we use Adam [17] optimizer. Learning rates of all models are chosen from $\{1e-2, 1e-3, 1e-4\}$. For all methods, we evaluate with rank $R \in \{3, 5, 8, 10\}$. All methods are evaluated by 5 runs with different random seeds.

---

[2] https://github.com/ctilling/SparseTensorHGP
[3] https://github.com/ctilling/POND
[4] https://github.com/USC-Melady/KDD19-CoSTCo

Table 2: Continuous-time tensor completion

| | RMSE | | | | MAE | | | |
|---|---|---|---|---|---|---|---|---|
| *Air* | Rank 3 | Rank 5 | Rank 8 | Rank 10 | Rank 3 | Rank 5 | Rank 8 | Rank 10 |
| CTCP | $1.020 \pm 0.002$ | $1.022 \pm 0.002$ | $1.022 \pm 0.002$ | $1.022 \pm 0.002$ | $0.784 \pm 0.002$ | $0.785 \pm 0.002$ | $0.787 \pm 0.002$ | $0.787 \pm 0.002$ |
| CTGP | $0.475 \pm 0.000$ | $0.463 \pm 0.000$ | $0.459 \pm 0.000$ | $0.458 \pm 0.000$ | $0.318 \pm 0.000$ | $0.304 \pm 0.000$ | $0.301 \pm 0.000$ | $0.299 \pm 0.000$ |
| CTNN | $1.013 \pm 0.001$ | $1.005 \pm 0.005$ | $0.999 \pm 0.009$ | $1.013 \pm 0.002$ | $0.780 \pm 0.001$ | $0.777 \pm 0.003$ | $0.776 \pm 0.003$ | $0.780 \pm 0.001$ |
| NNDTN | $0.377 \pm 0.004$ | $0.364 \pm 0.002$ | $0.334 \pm 0.004$ | $0.328 \pm 0.004$ | $0.247 \pm 0.003$ | $0.239 \pm 0.002$ | $0.217 \pm 0.003$ | $0.212 \pm 0.004$ |
| NONFAT | $0.339 \pm 0.002$ | $0.335 \pm 0.002$ | $0.351 \pm 0.005$ | $0.342 \pm 0.002$ | $0.224 \pm 0.002$ | $0.219 \pm 0.001$ | $0.228 \pm 0.003$ | $0.223 \pm 0.001$ |
| THIS-ODE | $0.569 \pm 0.001$ | $0.566 \pm 0.004$ | $0.542 \pm 0.005$ | $0.541 \pm 0.002$ | $0.415 \pm 0.002$ | $0.409 \pm 0.004$ | $0.395 \pm 0.004$ | $0.391 \pm 0.001$ |
| EnergyTD | $\mathbf{0.302 \pm 0.008}$ | $\mathbf{0.291 \pm 0.006}$ | $\mathbf{0.300 \pm 0.012}$ | $\mathbf{0.283 \pm 0.004}$ | $\mathbf{0.184 \pm 0.006}$ | $\mathbf{0.177 \pm 0.003}$ | $\mathbf{0.172 \pm 0.006}$ | $\mathbf{0.184 \pm 0.003}$ |
| *Click* | | | | | | | | |
| CTCP | $2.063 \pm 0.009$ | $2.020 \pm 0.025$ | $2.068 \pm 0.012$ | $2.009 \pm 0.023$ | $1.000 \pm 0.009$ | $0.977 \pm 0.021$ | $1.005 \pm 0.010$ | $0.969 \pm 0.012$ |
| CTGP | $1.424 \pm 0.002$ | $1.423 \pm 0.004$ | $1.404 \pm 0.004$ | $1.392 \pm 0.002$ | $0.880 \pm 0.003$ | $0.877 \pm 0.003$ | $0.856 \pm 0.002$ | $0.849 \pm 0.001$ |
| CTNN | $1.820 \pm 0.005$ | $1.820 \pm 0.005$ | $1.820 \pm 0.005$ | $1.820 \pm 0.005$ | $1.077 \pm 0.027$ | $1.053 \pm 0.012$ | $1.083 \pm 0.016$ | $1.071 \pm 0.024$ |
| NNDTN | $1.418 \pm 0.005$ | $1.409 \pm 0.004$ | $1.407 \pm 0.002$ | $1.410 \pm 0.004$ | $0.858 \pm 0.002$ | $0.856 \pm 0.002$ | $0.859 \pm 0.003$ | $0.863 \pm 0.006$ |
| NONFAT | $1.400 \pm 0.008$ | $1.411 \pm 0.006$ | $1.365 \pm 0.004$ | $\mathbf{1.351 \pm 0.002}$ | $0.853 \pm 0.004$ | $0.873 \pm 0.004$ | $0.832 \pm 0.004$ | $0.812 \pm 0.002$ |
| THIS-ODE | $1.421 \pm 0.004$ | $1.413 \pm 0.002$ | $1.408 \pm 0.002$ | $1.395 \pm 0.003$ | $0.836 \pm 0.004$ | $0.836 \pm 0.003$ | $0.832 \pm 0.002$ | $0.829 \pm 0.003$ |
| EnergyTD | $\mathbf{1.396 \pm 0.003}$ | $\mathbf{1.385 \pm 0.003}$ | $\mathbf{1.356 \pm 0.001}$ | $1.357 \pm 0.001$ | $\mathbf{0.777 \pm 0.003}$ | $\mathbf{0.775 \pm 0.003}$ | $\mathbf{0.772 \pm 0.002}$ | $\mathbf{0.773 \pm 0.001}$ |

The completion results are shown in Table 1. The mean and standard deviation of the root mean square error (RMSE) and mean absolute error (MAE) are reported. Results from POND are not included for ACC due to the slow code execution and the substantial memory requirements for this dataset. Our model outperforms the baselines in nearly all cases on both RMSE and MAE. Moreover, the improvement is significant ($p < 0.05$) for most cases. All non-linear methods exhibit substantial superiority over CP-WOPT, thereby highlighting the advantage of employing more flexible structures. Furthermore, the enhanced performance of our model may be attributed to the utilization of more flexible undirected probabilistic distributions. It is worth noting that our model, despite employing MLP layers, outperforms POND and CoSTCo, which utilize convolutional layers. We believe that our model can be further improved by designing more suitable network architectures.

### 5.2.2 Continuous-time tensor completion

In this subsection, we evaluate our model on two continuous-time tensor datasets: (1) *Air*, the Beijing air quality dataset [47] of shape *12 sites $\times$ 6 pollutants* with about $1 \times 10^4$ observations at different time stamps; (2) *Click*, an ads click dataset [43] of shape *7 banner positions $\times$ 2842 site domains $\times$ 4127 mobile APPs* with about $5 \times 10^4$ entries at different time stamps. We use the same dataset split as in [43] and report the 5-fold cross validation results.

**Competing methods** We compare with (1) Nonparametric factor trajectory learning [NONFAT, 43], which is the SOTA method for continuous-time tensor completion, and other baselines including: (2) Continuous-time CP [CTCP, 48], which uses polynomial splines to model dynamics of CP coefficients; (3) Continuous-time GP (CTGP), which extends GPTF [53] by injecting time stamps into GP kernels; (5) Continuous-time NN decomposition (CTNN), which directly uses time stamps as inputs in CoSTCo [25] to learn continuous dynamics; (5) Discrete-time NN decomposition with non-linear dynamics [NNDTN, 43], which employs RNN dynamics for time steps; (6) Tensor high-order interaction learning via ODEs [THIS-ODE, 24], where continuous-time trajectories of tensor entries are captured by neural ODEs. For baseline models (1-5), we use the official implementation[5] provided by [43]. For (6) THIS-ODE, its official implementation[6] is used.

**Experimental settings and results** We set batch size 128 and run 400 epochs for *Air*, 200 epochs for *Click*. Notably, in the case of THIS-ODE, we only run 200 epochs for *Air* and 75 epochs for *Click* due to the slow speed of ODE solvers (See Appendix C.3 for an illustration). For our model, we use Adam [17] optimizer with learning rate chosen from $\{1e-2, 1e-3\}$. The network architecture is described in Section 3.1. For baseline models, provided settings in their codebases are used. For all methods, we evaluate with rank $R \in \{3, 5, 8, 10\}$. All methods are evaluated by 5 runs with different random seeds.

---

[5] https://github.com/wzhut/NONFAT
[6] https://github.com/shib0li/THIS-ODE

Table 2 presents the completion results. It should be noted that the results presented here differ from those reported in [43] as we adhere to the standard definition of RMSE and MAE (see appendix for detail). Our model surpasses the baseline methods in almost all cases for both RMSE and MAE, with statistically significant improvements ($p < 0.05$) observed in most cases. Particularly, we observe that the improvements in MAE are notably more significant. One possible reason is that NONFAT is trained implicitly by minimizing the square loss (with regularization) as it adopts Gaussian assumption about the data. However, our model does not make such assumptions about the data distribution and the loss function. Hence, it can adapt to the data distribution more flexibly. Additionally, we observe that directly injecting time stamps into neural networks, as done in CTNN, is ineffective, thus highlighting the advantage of our model in learning more informative structures.

## 6   Conclusion

We introduce an innovative approach to undirected probabilistic tensor decomposition (TD), characterized by its exceptional flexibility in accommodating various structures and distributions. Specifically, our model integrates deep EBMs in TD to relax both structural and distributional assumptions, enabling it to handle complex real-world applications. To efficiently learn the doubly intractable pdf, we derive a VCNCE objective that is the upper bound of the CNCE loss. Experimental results demonstrate that our model can handle diverse distributions and outperforms baseline methods in multiple real-world applications. One limitation is that our final loss function is not a fully variational upper bound of CNCE, since we have to use importance samples to approximate the pdf of noise samples in Eq. (7). In the future, we aim to derive a fully variational bound as in [46]. Finally, we did not delve into the interpretability of learned factors in this work. However, exploring the interpretability of these factors represents a promising avenue for future research in the realm of tensor decompositions.

## Acknowledgments

Zerui Tao was supported by the RIKEN Junior Research Associate Program. This work was supported by the JSPS KAKENHI Grant Numbers JP20H04249, JP23H03419.

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

# A Proof of Theorem 1

Firstly, we give the definition of $f$-divergence.

**Definition 1** ($f$**-divergence**) *The $f$-divergence between two probability density functions (pdf) $p$ and $q$ is defined as,*

$$\mathbb{D}_f(p\|q) = \mathbb{E}_q\left[f\left(\frac{p}{q}\right)\right],$$

*where $f : [0,\infty) \to \mathbb{R}$ is a convex function and $f(1) = 0$.*

As shown in [33], since partition functions for $\phi(x, m; \theta)$ and $\phi(x; \theta)$ are the same, we have the following factorization,

$$\phi(x, \boldsymbol{m}; \theta) = \phi(x; \theta)p(\boldsymbol{m} \mid x; \theta).$$

The difference between the two objective becomes,

$$
\begin{aligned}
&\mathcal{L}_{\text{VCNCE}}(\theta, \varphi) - \mathcal{L}_{\text{CNCE}}(\theta) \\
=&2\mathbb{E}_{xy}\mathbb{E}_{q(\boldsymbol{m};\varphi)}\left\{\log\left[1 + \frac{\phi(y;\theta)p_c(x \mid y)q(\boldsymbol{m};\varphi)}{\phi(x,\boldsymbol{m};\theta)p_c(y \mid x)}\right] - \log\left[1 + \frac{\phi(y;\theta)p_c(x \mid y)}{\phi(x;\theta)p_c(y \mid x)}\right]\right\} \\
=&2\mathbb{E}_{xy}\mathbb{E}_{q(\boldsymbol{m};\varphi)}\log\frac{\phi(x,\boldsymbol{m};\theta)\phi(x;\theta)p_c(y \mid x) + \phi(y;\theta)p_c(x \mid y)\phi(x;\theta)q(\boldsymbol{m};\varphi)}{\phi(x,\boldsymbol{m};\theta)\phi(x;\theta)p_c(y \mid x) + \phi(y;\theta)p_c(x \mid y)\phi(x,\boldsymbol{m};\theta)} \\
=&2\mathbb{E}_{xy}\mathbb{E}_{q(\boldsymbol{m};\varphi)}\log\frac{p(\boldsymbol{m} \mid x;\theta)\phi(x;\theta)p_c(y \mid x) + \phi(y;\theta)p_c(x \mid y)q(\boldsymbol{m};\varphi)}{p(\boldsymbol{m} \mid x;\theta)\phi(x;\theta)p_c(y \mid x) + \phi(y;\theta)p_c(x \mid y)p(\boldsymbol{m} \mid x;\theta)} \\
=&2\mathbb{E}_{xy}[\mathbb{D}_{f_{xy}}(p(\boldsymbol{m} \mid x;\theta)\|q(\boldsymbol{m}))],
\end{aligned}
$$

where

$$f_{xy}(u) = \log\left(\frac{\kappa_{xy} + u^{-1}}{\kappa_{xy} + 1}\right),$$

with $\kappa_{xy} = \frac{\phi(x;\theta)p_c(y|x)}{\phi(y;\theta)p_c(x|y)}$. It is straightforward to verify that $f(1) = 0$. The derivatives of $f$ is

$$f'(u) = -\frac{1}{u^2\kappa + u}, \quad f''(u) = \frac{2u\kappa + 1}{(u^2\kappa + u)^2}.$$

Since $\kappa$ and $u$ are positive, $f$ is a convex function. Therefore, $f$ satisfy the requirements of $f$-divergence.

# B Proof of Corollaries 1 and 2

Corollary 1 is a straightforward consequence of Theorem 1. Since the $f$-divergence becomes zero if and only if the two distributions are identical, we have,

$$\mathcal{L}_{\text{VCNCE}}(\theta, \varphi) = \mathcal{L}_{\text{CNCE}}(\theta) \iff q(\boldsymbol{m}; \varphi) = p(\boldsymbol{m} \mid x; \theta).$$

Moreover, since the $f$-divergence is positive and Theorem 1, we have

$$p(\boldsymbol{m} \mid x; \theta) = \underset{q(\boldsymbol{m};\varphi)}{\arg\min}\,\mathcal{L}_{\text{VCNCE}}(\theta, q(\boldsymbol{m};\varphi)).$$

Then, plugging the optimal distribution gives the tight bound, we have,

$$\min_\theta \mathcal{L}_{\text{CNCE}}(\theta) = \min_\theta \min_{q(\boldsymbol{m};\varphi)} \mathcal{L}_{\text{VCNCE}}(\theta, \varphi).$$

# C Experimental details

## C.1 Simulation study

**Tensors with non-Gaussian distributions** For both GPTF and our model, we set batch size to 1000 and run 500 epochs with Adam optimizer. The initial learning rate is $1e-3$ and subsequently reduced

by 0.3 at $60\%, 75\%$ and $90\%$ of the maximum epochs. Moreover, the rank is set to 3 for both models. For GPTF, radial basis function (RBF) kernel with band width 1.0 is used, where 100 inducing points is adopted for approximation. For the conditional distribution $p(x_\mathbf{i} \mid \boldsymbol{m}_\mathbf{i}) = \mathcal{N}(x_\mathbf{i} \mid f(\boldsymbol{m}_\mathbf{i}), \sigma^2)$ in GPTF, $\sigma$ is fixed and chosen as the sample standard variance. For our model, we use 5 hidden layers of width 64 for both $g_1$, $g_3$ and $g_4$ defined in Section 3. $g_2$ is a summation layer. We use ELU activation for non-linearity. For the VCNCE loss, the conditional noise distribution is set as $p_c(y \mid x) = \mathcal{N}(y \mid x, 0.3^2)$ and $\nu = 10$ noise samples are used for each data point.

**Continuous-time tensors**  The data sizes and optimization parameters are the same with the previous simulation. The rank of all models are set to 3. For NONFAT, 100 inducing points are used to approximate the kernel function. We run the NONFAT model for 5000 epochs because we find that the algorithm converges very slowly. Other hyper-parameters are chosen by their default settings. For BCTT, we do not modify their code and settings. For our model, we use 3 hidden layers of length 64 with ELU activation. The conditional noise distribution in the VCNCE loss is set to $p_c(y \mid x) = \mathcal{N}(y \mid x, 1)$ and $\nu = 20$ noise samples are used for each datum.

## C.2 Tensor completion

For all datasets, when training our model, we scale the data to $[0, 1]$ based on the *training* data. For testing, we multiply the scale statistic computed by the training data and evaluate the performance on the original domain. We do not employ such data normalization for baselines models, because that will influence their default settings.

### C.2.1 Sparse tensor completion

For both *Alog* and *ACC*, the batch size is set to 1000. We run 1000 epochs for *Alog* and *100* epochs for *ACC* due to their different sample numbers. For *Alog* dataset, we add i.i.d. Gaussian noises from $\mathcal{N}(0, 0.05^2)$ during training, while for *ACC*, the standard variance is set to $0.02$. The Adam optimizer is used with learning rate chosen from $\{1\mathrm{e}{-}2, 1\mathrm{e}{-}3, 1\mathrm{e}{-}4\}$. We also use gradient clip with maximum infinity norm of 2.0 for training stability. Moreover, we use learning rate scheduler by reducing the initial learning rate by 0.3 at 40%, 60%, and 80% of the total iterations. For both datasets, we use 2 hidden layers of length 50 with ELU activation for $g_1$, $g_3$ and $g_4$ for our model. For the VCNCE loss, we set $\nu = 20$ noise samples with noise variance tuned from $\{0.3^2, 0.5^2, 0.8^2, 1.0^2\}$. In practice, we find that the noise variance is influential to the final performance, even we are using conditional noises. However, with VCNCE, there is only one hyper-parameter for the noise distribution. While for CNCE, one may need to tune both mean and variance of the noise.

### C.2.2 Continuous-time tensor completion

For *Air* and *Click* datasets, we set batch size to 128. We run 400 epochs for *Air* and 200 epochs for *Click* due to their different data sizes. For *Alog* dataset, we add i.i.d. Gaussian noises from $\mathcal{N}(0, 0.05^2)$ during training, while for *ACC*, the variance is set as $0.15^2$. To encode the temporal information into the energy function, we use the sinusoidal positional encoding, as described in Section 3. Other settings are the same with Appendix C.2.1.

It should be noted that we use the standard definition of root mean square error (RMSE) and mean absolute error (MAE), namely,

$$\mathrm{RMSE} = \sqrt{\frac{\sum_{i=1}^{N}(x_i - \hat{x}_i)^2}{N}}, \quad \mathrm{MAE} = \frac{\sum_{i=1}^{N}|x_i - \hat{x}_i|}{N},$$

where $x_i$ is the ground truth and $\hat{x}_i$ is the estimate. Therefore, the results are different from those presented in [43], where the authors used *relative* versions of RMSE and MAE,

$$\mathrm{RMSE} = \sqrt{\sum_{i=1}^{N} \frac{(x_i - \hat{x}_i)^2}{x_i^2}}, \quad \mathrm{MAE} = \sum_{i=1}^{N} \frac{|x_i - \hat{x}_i|}{|x_i|}.$$

We modify the evaluation part of their code[7] and report the results.

---

[7] https://github.com/wzhut/NONFAT

## C.3 Computational time

The time complexity of our model is $\nu$ times greater than traditional TDs, since we need to compute forward passes for $\nu$ particles. However, as we only use small networks, the computational speed is still very fast. To illustrate, we compare the runtime performance of several baselines and our model on a single RTX A5000 GPU. We conduct tests on the *Air* dataset, with a batch size of 128 and tensor rank of 5. The reported running time is the average of the first 10 epochs. For our model, we set $\nu = 20$. The default settings are used for other baselines. Table 3 lists the computational time of CTGP, NNDTN, NONFAT and THIS-ODE, all of which perform better than other baselines. The results show that our model achieves better performances within a reasonable time, especially compared to THIS-ODE.

Table 3: Computing time

|  | CTGP | NNDTN | NONFAT | THIS-ODE | EnergyTD |
|---|---|---|---|---|---|
| Time/Epoch (in seconds) | 1.17±0.30 | 2.18±0.04 | 2.51±0.13 | 464.±131. | 5.30±0.37 |

## C.4 Ablation study on the objective function

We conduct an additional ablation study to show the advantage of VCNCE over the variational noise-contrastive estimation [VNCE, 33] objective. The main difference between the VNCE and VCNCE is that VNCE uses noises from a fixed Gaussian distribution, *e.g.*, $y \sim p_n(y) = \mathcal{N}(y \mid \mu, \sigma^2)$, while VCNCE uses conditional noises, *e.g.*, $y \sim p_c(y \mid x) = \mathcal{N}(y \mid x, \sigma^2)$. Hence, these two strategies yield different objective functions. The objective function of VNCE is defined as

$$\mathcal{L}_{\text{VNCE}} = \mathbb{E}_x \mathbb{E}_{q(\boldsymbol{m}|x;\varphi)} \log \left( \frac{\phi(x, \boldsymbol{m}; \theta)}{\phi(x, \boldsymbol{m}; \theta) + \nu q(\boldsymbol{m} \mid x; \varphi) p_n(x)} \right)$$
$$+ \nu \mathbb{E}_y \log \left( \frac{\nu p_n(y)}{\nu p_n(y) + \mathbb{E}_{q(\boldsymbol{m}|y)} \left[ \frac{\phi(y, \boldsymbol{m}; \theta)}{q(\boldsymbol{m}|y)} \right]} \right),$$

where $p_n(\cdot)$ is the fixed noise distribution. For VNCE, choosing inappropriate noise distributions may result in bad performances.

We test the proposed model on the *Air* dataset, training on the VCNCE loss and VNCE loss, respectively. We set the batch size to 128 and run 400 epochs. Adam optimizer with initial learning rate $1\mathrm{e}{-2}$ is adopted. The initial learning rate is subsequently reduce by 0.3 at $20\%, 50\%$ and $80\%$ of the total epochs. For VNCE, we set $\mu = 0$, which is a common practice in relevant literature. To show how the noise variance affects the learning process, we test different noise variances, *e.g.*, $\sigma \in \{0.3, 0.5, 0.7\}$ for both VNCE and VCNCE. Other settings are the same with Appendix C.2.2.

Fig. 3 depicts the RMSE and MAE on the test data when optimizing VNCE and VCNCE objective functions. We test five runs, plot mean values in lines and standard deviations in shadowed areas. It is shown that VCNCE gets better and more stable results on both RMSE and MAE.

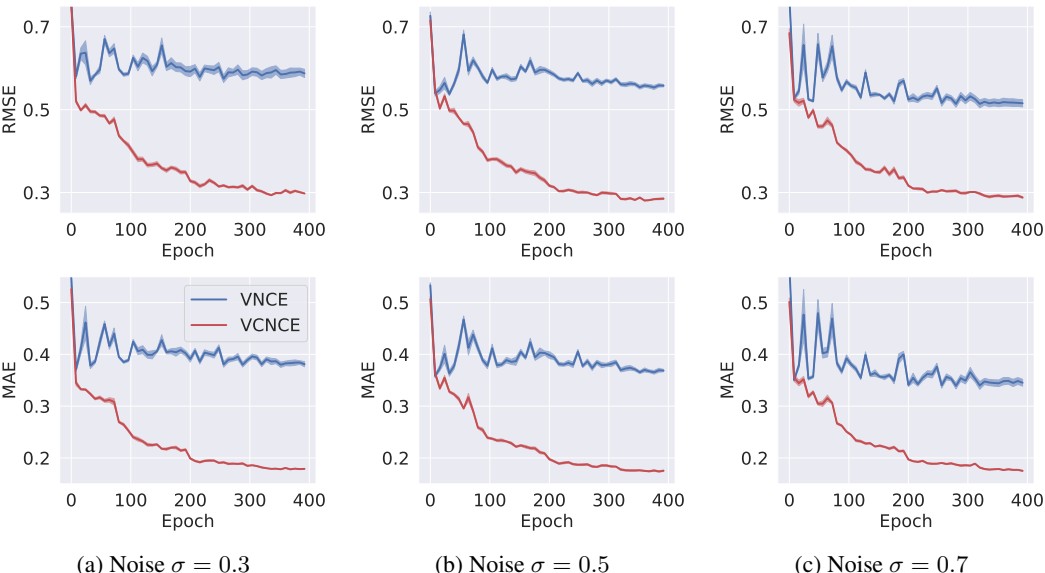

Figure 3: Learning process of optimizing the VNCE and VCNCE loss. The first row is RMSE and the second row is MAE.

