# OpenReview forum: "Undirected Probabilistic Model for Tensor Decomposition"
_NeurIPS.cc/2023/Conference — NeurIPS 2023 poster_

### Official Review · Reviewer_7dHC · 2023-06-18

**Soundness:** 3 good
**Presentation:** 2 fair
**Contribution:** 3 good
**Rating:** 6
**Confidence:** 2

**Summary:**

The authors propose a new Probabilistic Tensor Decomposition (TD) method. By modeling the joint probability of the data and latent tensor factors using an Energy Based approach, they make no (possibly restrictive) structural and distributional assumptions on the generative process linking latent and observations. The model is trained using an upper bound of the Conditional Noise-Contrastive Estimation Loss. It is then evaluated on simulations using non Gaussian generative models and on tensor completion task on real world datasets.


**Strengths:**

The possibility to bypass the specification of both structural and distributional dependencies is very interesting. It has many applications beyond the tensor completion tasks, for example when analyzing datasets with complicated dependencies on significantly simpler latent variables.

The simulated datasets used in the paper are simple but illustrate this point well.

Results on real world tensor completion tasks are convincing.

**Weaknesses:**

Major Points:

1) The code is not provided.

2) In the introduction, it would be worth motivating the use of probabilistic methods over traditional ones.

3) I did not find the Energy Based Model section (2.2) very well written. In particular, it is not clear how to choose the conditional $p_c(y|x)$  to improve efficiency.

4) The method is only evaluated on tensor completion tasks. Do the inferred factors have any explanatory value ? How robust is the decomposition ?

5) There is no discussion on the compute time of the proposed method.

Minor Points:

1) l.43 It is worth mentioning that Generalized CP decomposition exist but that most of them do not allow a probabilistic treatment of the data. Moreover, beyond Gaussian and Binomial, tractable decomposition have been developed for Negative Binomial Distributions.

2) l.93 I think the authors implicitly assume $R_d = R$ for all $d$ without explicitly saying it.

3) Experimental details in the main paper and supplementary are not consistent (experiment 1: main paper $R=5$, supplementary $R=3$).


**Questions:**

1) The paper only discusses 2 and 3-way tensors. How does the method scale with higher order tensors ?

2) Beside Tensor completion, an interesting aspect of TD methods is often to explain a dataset $\mathcal{X}$ using simpler yet unobserved factors $\mathcal{Z}$. How robust are the inferred factors depending on the initialization of the model ? Can you measure it using e.g. a similarity metric and compare it to existing methods ?

3) The authors argue that the use of overly simple generative model $p(x|z)$ can bias the TD decomposition. They provide convincing illustration on their toydataset but it would be interesting to better understand why it is the case on the tested the real world dataset.

4) The approach developed by the authors seems extremely flexible. Yet, by allowing almost arbitrary complex link between the latent and the observation, one might fear that the inferred latent distribution is "non-unique", which might hinder the interpretation of the discovered factors. Can you say more about identifiability ?

**Limitations:**

There is no discussion on the limitation of the method when the posterior $p(z|x)$ itself is non Gaussian. The approximation $q(m; \phi)$ is set to be multivariate normal with diagonal covariance, and, so, it cannot model, for example, multi-modal distribution on $z$.

---

> ### Author Rebuttal · Authors · 2023-08-09
>
> ## W1: Code availability.
> We send an anonymous link to the AC via the official comment.
>
> ## W2: Motivation of using probabilistic methods over traditional ones.
> There are several advantages of using probabilistic models, such as,
>
> 1. Using probabilistic models, we can deal with different data types.
> For examples, by adopting different distributions, it provides a principled way
> of designing proper loss functions.
> 2. Probabilistic models can give uncertainty estimates about the observations and latent factors.
> 3. Bayesian models have other potential benefits, such as continual learning, less likely to over-fit and so on.
>
> We will add more discussions about this part in the introduction.
>
> ## W3: How to choose conditional noises to improve efficiency.
>
> The basic idea of NCE is to identify the noises from the data points, which shares some similarities with density ratio estimation such as in GAN. Unfortunately, there is no theoretical guidance about how to choose noise distributions yet. A common belief is that the noises should be similar to data. This is the motivation of using data-dependent noises, rather than the same noise for each data point. We will rearrange the presentation to make it more clear.
>
> ## W4: Explanatory value of factors? How robust is the decomposition?
>
> Currently, we do not focus on learning explanatory factors. To learn such factors, one may need to add additional constraints on the tensor decomposition, e.g., identifiability of the factors. This is generally hard for non-linear models using neural networks. However, we conduct some simple investigation on the *Alog* dataset following the setting of [1]. In specific, the tensor latent factors are firstly mapped to the two-dimensional plane through PCA and then the $k$-means clustering results are plotted for each data point. The figure can be found in the attached PDF. We find hidden patterns similar to Figure 4 in [1]. Moreover, we investigate the results for different ranks and initializations, showing that this pattern is not due to random effects. Although we initially did not consider this problem, our model seems to have better clustering results compared with other baselines. This indicates the potential of our model to discover explanatory latent factors.
>
> Currently, we do not conduct experiments about robustness. It would be a very interesting direction since our model is capable of handling different data distributions. Moreover, the adversarial robustness would also be an interesting direction to investigate.
>
> [1]. Tillinghast, C., Wang, Z., & Zhe, S. (2022). Nonparametric sparse tensor factorization with hierarchical Gamma processes. In ICML.
>
> ## W5: Compute time.
>
> Due to the space limit, please check our response to *W1* raised by **Reviewer Duah*.
>
> ## Minor1: Tractable tensor decomposition for other distributions
>
> We agree that there are many tractable TDs for other distributions. We will add several references about this point, such as [1-3].
>
> [1]. Schein, A., Zhou, M., Blei, D. & Wallach, H.. (2016). Bayesian Poisson Tucker Decomposition for Learning the Structure of International Relations. In ICML.
>
> [2]. Hong, D., Kolda, T. G., & Duersch, J. A. (2020). Generalized canonical polyadic tensor decomposition. SIAM Review.
>
> [3]. Soulat, H., Keshavarzi, S., Margrie, T., & Sahani, M. (2021). Probabilistic tensor decomposition of neural population spiking activity. In NeurIPS.
>
> ## Minor2: Assume $R_d = R$.
>
> We will highlight this assumption.
>
> ## Minor3: Inconsistent setting of rank.
>
> We checked experimental records and the rank should be 3. We will fix it.
>
> ## Q1: Scalability with higher order tensors
>
> In principle, both the time and space complexity of our model is linear with the tensor order $D$. Given the same number of observations $N$, our model is scalable with the tensor order $D$.
>
> ## Q2: Robustness of the learning process of latent factors.
>
> Generally, we think it is hard to learn  identifiable latent factors for non-linear models, especially using neural networks. However, we do find robustness of these latent factors in some sense. For example, in our response to *W4*,
> we plot the latent factors for different ranks and different initializations.
> They show similar patterns in the clustering.
> This would be an interesting direction to study further, including new metrics to evaluate these similarities.
>
> ## Q3: Different distributions in real applications.
>
> Demonstrating this phenomenon is hard in real applications, since we do not know the underlying real distributions. However, we can find some evidence from current results. In Table 1 of the manuscript and the additional results in our response to **Reviewer buha**, we observe that the improvements in MAE are notably more significant. One possible reason is that other baselines are trained implicitly by minimizing the square loss as they adopt Gaussian assumptions about the data. However, our model does not make such assumptions about the data and the loss function. Hence, it can adapt to the data distribution more flexibly. Additionally, in Table 2 of manuscript, while simply using NN to model the trajectory is hard, as shown by NNDTL and CTNN, our model outperforms them and NONFAT, which is the SOTA in the field. This may indicate that our model efficiently learns how the data distribution shifts with time.
>
> ## Q4: Identifiability about the tensor factors.
>
> Unfortunately, we think the model cannot learn identifiable latent factors currently, since we are using non-linear neural networks. However, as our response to *W4*, our model possibly learns some sharing latent patterns under different settings and different initializations.
>
> ## Limitation: Non-Gaussian posteriors.
>
> We agree with the reviewer about this limitation. However, due to the computational efficiency, mean-field Gaussian assumption is widely used in variational inference. It could be a potential direction to improve our model, by using more expressive distributions.

---

> > ### Comment · Reviewer_7dHC · 2023-08-11
> >
> > Thank for your response.
> >
> > Although I appreciate the effort of answering my concerns/questions, the results provided for W4 - Q2  are not satisfying. The authors should either (i) carefully evaluate the robustness of their model or (ii) clearly state that interpretability and robustness of the discovered factors is beyond the scope of the study, which only evaluates the method for Sparse Tensor Completion (and which is fine...)
> >
> > In its current state:
> >
> > (i) The Alog dataset is not described.
> > (ii) The authors mentioned "different ranks and initializations" but only 1 seed per rank is plotted on the attached .pdf.
> > (iii) No interpretation on the discovered cluster is provided
> > (iv) The author do not report any robustness metric (see for example [1]) and instead use vague statements like "our model seems to have better clustering results compared with other baselines".
> >
> > [1] Williams et al., 2018, Neuron 98

---

> > > ### Author Response · Authors · 2023-08-11
> > >
> > > Thanks very much for your comments and providing the reference.
> > > We apologize for any ambiguity raised by the previous response.
> > > We should clarify that in this work we do not aim to study the explanatory value of the tensor factors nor the robustness of the decomposition.
> > > And we will not make such claims without further theoretical or experimental results.
> > > The purpose of the previous response is to show some potential evidence for future research, which we think would be interesting.
> > >
> > > Regarding the questions
> > >
> > > (i). The Alog dataset is not described.
> > >
> > > >The Alog dataset is an order-3 tensor, extracted from an access log of a file management system. Each dimension represents *user*, *action*, *resource*. We included the above descriptions in the manuscript. This dataset was initially processed by [1], and has been widely used to evaluate completion performances, such as in [2-4], which are also our baseline models. However, we can hardly find the original data and detailed description about each dimension, since only processed data were provided. For example, mode-2 represents 100 actions, but we do not know what these actions are.  So we were not able to explain the clustering results. The clustering results were presented following [4] to show some preliminary evidence.
> > > >-  [1]. Zhe, S., et al. Scalable nonparametric multiway data analysis. AISTATS (2015).
> > > >- [2]. Zhe, S., et al. Distributed flexible nonlinear tensor factorization. NeurIPS (2016).
> > > >- [3]. Tillinghast, C., et al. Probabilistic neural-kernel tensor decomposition. ICDM (2020).
> > > >- [4]. Tillinghast, C., et al. Nonparametric sparse tensor factorization with hierarchical Gamma processes. ICML (2022).
> > >
> > > (ii). The authors mentioned "different ranks and initializations" but only 1 seed per rank is plotted on the attached .pdf
> > >
> > > >We are sorry about the ambiguity raised by this sentence. Here, “different ranks and initializations” means that the models with different ranks were initialized differently from each other.
> > >
> > > (iii). No interpretation on the discovered cluster is provided
> > >
> > > >As we described in the answer to question (i), since this dataset is mainly used to evaluate completion performances and the original data resources were not released, we were not able to give detailed interpretation about the clustering results, for example, the exact meaning of each cluster.
> > >
> > > (iv). The author do not report any robustness metric (see for example [1]) and instead use vague statements like "our model seems to have better clustering results compared with other baselines".
> > >
> > > >Thanks for providing the reference. Currently, we do not conduct numerical evaluation about the robustness of the decomposition nor the interpretation of the tensor factors. However, since we focus on tensor completion, the prediction results of those missing entries are evaluated under different initializations and data folds. Maybe datasets from neural science etc. are more suitable for explaining the tensor factors, where we can use domain knowledge for explanation. We will leave this part for future research.
> > >
> > > If there are any unclear issues in our response or if you have any further questions, we are more than happy to answer them.

---

> > > > ### Comment · Reviewer_7dHC · 2023-08-14
> > > >
> > > > Thank you for your answer. I confirm that clearly stating that interpretability and robustness of the discovered factors is beyond the scope of the study is probably best then.

---

> > > > > ### Author Response · Authors · 2023-08-14
> > > > >
> > > > > Thanks very much for your constructive comments that are inspiring for our research!
> > > > > We will clearly add discussions about this issue in the manuscript.

---

### Official Review · Reviewer_Duah · 2023-07-01

**Soundness:** 4 excellent
**Presentation:** 4 excellent
**Contribution:** 4 excellent
**Rating:** 6
**Confidence:** 4

**Summary:**

This paper proposes a probabilistic tensor decomposition model called EnergyTD that integrates a deep energy-based model (EBM) in tensor decomposition. The EnergyTD does not model the values in a tensor as the conditional probability conditioned on the latent factors based on the predefined models but models them as the joint probability of the data and the latent factors. It provides a learning algorithm that extends the CNCE loss using the variational approach. The experimental results show that EnergyTD outperforms conventional methods in the tensor decomposition tasks assuming diverse distributions including continuous-time tensors, tensor completion, and continuous-time tensor completion.

**Strengths:**

This paper proposes a probabilistic tensor decomposition model that integrates EBM in tensor decomposition along with a theoretical guarantee. The experimental results show that the proposed method outperforms conventional methods in the tensor decomposition tasks assuming diverse distributions. This paper is well-organized and easy to follow.

**Weaknesses:**

The time/space complexity of the proposed method is only poorly discussed, and there is no evaluation regarding the computational cost. This is one of the major concerns in the tensor decomposition research community.

A minor problem: The "difficultly" in line 109 should be "difficulty".

**Questions:**

What is the time/space complexity of the proposed method? How is the proposed method computationally effective compared to the conventional methods?

**Limitations:**

As the authors describe in the Conclusion, there is room to improve theoretical analysis. Moreover, the architecture of the network structure should be discussed more in the future.

---

> ### Author Rebuttal · Authors · 2023-08-09
>
> ## W1: Time/space complexity.
>
> The time complexity of training should be $\mathcal{O}(\nu B ( D R H + L H^2))$, where $B$ is the batch size, $\nu$ is the number of conditional noises, $H$ is the number of hidden units per layer, $L$ is the number of layers and $D$ is the tensor order. The space complexity should be $\mathcal{O}(D R + LH^2)$ to store tensor factors and NN parameters. The space complexity is linear to the tensor order and rank, which is similar to many classical TDs such as CP and tensor train decomposition. We additionally use an NN. However, the network is small, e.g., several MLP layers with hundreds of hidden units each. The time complexity is $\nu$ times more than traditional ones, since we need to compute forward passes for $\nu$ particles. However, since these forward passes can be computed in parallel, we believe the computational speed can be easily improved using parallel programming libraries such as `JAX` and `functorch`. Moreover, unlike many traditional TDs performing ALS updates on the whole datasets, we use stochastic optimization, which is highly scalable to large datasets.
>
> To illustrate the computing time, we post an example of training the *Air* dataset, with rank 5. We compare with CTGP, NNDTN, NONFAT, which are better than other baselines, and THIS-ODE suggested by **Reviewer buha**. All experiments are conducted on a single RTX A5000 GPU. For more details, please check our response to **Reviewer buha**.
>
> |                         | CTGP            | NNDTN           | NONFAT          | THIS-ODE        | EnergyTD        |
> | ----------------------- | --------------- | --------------- | --------------- | --------------- | --------------- |
> | Time/Epoch (in seconds) | 1.17 $\pm$ 0.30 | 2.18 $\pm$ 0.04 | 2.51 $\pm$ 0.13 | 464. $\pm$ 131. | 5.30 $\pm$ 0.37 |
>
> Our model is slower than NONFAT, but much faster than THIS-ODE. While NONFAT is faster for each epoch, it converges much slower than our model. As suggested by the paper of NONFAT, we run 10,000 epochs to get a good performance. For our model, only 200 epochs is sufficient to converge.
>
>
> ## W2: Typo
>
> Thanks for pointing out the typo. We will fix it.
>
>
> ## Q1: What is the time/space complexity?
>
> Please see our response for *W1*.

---

> > ### Author Response · Authors · 2023-08-16
> >
> > Dear Reviewer Duah,
> >
> > Could we kindly know if the responses have addressed your concerns and if further explanations or clarifications are needed? Your time and efforts in evaluating our work is much appreciated.

---

> > ### Comment · Reviewer_Duah · 2023-08-21
> > **Thank you for your rebuttal!**
> >
> > Thank you very much for your answer to my question! I am satisfied with the discussion on computational complexity and I change my rating. I sincerely apologize for my late response.

---

> > > ### Author Response · Authors · 2023-08-21
> > >
> > > We are glad that we could address your concerns. Thank you for reviewing our work!

---

### Official Review · Reviewer_32MU · 2023-07-03

**Soundness:** 3 good
**Presentation:** 3 good
**Contribution:** 2 fair
**Rating:** 6
**Confidence:** 3

**Summary:**

This paper uses the Energy Based Model (EBM) framework to capture the joint probability of the data and latent tensor factors to learn as much information from data as possible, which discards the structural and distributional assumptions and thus avoid picking an inappropriate TD model.  To further flexibly learn the unnormalized probability function, the authors derive a variational upper bound of the conditional noise-contrastive estimation objective.  The main contribution lies in inserting the EBM framework into modeling probabilistic Tensor Decompositions.

**Strengths:**

Pros:

1. This paper first introduces undirect graphic models into modeling probabilistic tensor decomposition. In detail, the authors consider latent factor matrics as latent factors in the Variational Noise Contrastive Estimation framework, leading to flexible modeling of joint distribution over observations. To the best of my knowledge, this is interesting and novel to the Tensor Decomposition community.

2. This paper is well-written and the technological parts are sound and valid.

**Weaknesses:**

Cons:

1. If what I understand is correct,  one important goal of this paper is to find the $\theta$ and $p_{\theta}(x)$. The authors use the framework of variational noise contrastive estimation to obtain the unnormalized function $\phi(x;\theta)$. However, it seems to be easier to directly model $\phi(x;\theta)$ and thus minimize $\mathcal{L}_{CNCE}(\theta)$. Furthermore, it seems that these $\textbf{m}$s are unnecessary and can have arbitrary dimensions.

2. Most technical contributions are in the form of a combination of existing theory and framework. This actually decreases this paper's contributions.

**Questions:**

Questions:

1. As mentioned in the last part, I am confused about why not directly model $\phi(x;\theta)$ and $p_{\theta}(x)$.

**Limitations:**

Yes, the authors have adequately addressed the limitations and potential negative societal impact of their work.

---

> ### Author Rebuttal · Authors · 2023-08-09
>
> ## W1: Directly modeling $\phi(x; \theta)$ and the choice of $m$.
>
> Thanks for the question. Apart from the data distribution $\phi(x; \theta)$, in tensor decompositions, we aim to infer the latent factors, i.e., $z$ in the manuscript. If we directly model $\phi(x; \theta)$, we cannot obtain the information of these latent factors. Therefore, we need to model the joint energy function $\phi(x, m; \theta)$, where $m$ consists of $z$. Equipped with this joint energy function, we can sample posterior of the latent factors from $p(z \mid x)$. However, the data energy function $\phi(x; \theta) = \int \phi(x, m; \theta) dm$ becomes intractable, since we adopt non-linear mappings. This is one main difficulty in the learning process. Variational inference (VI) has become one of the most popular tools to deal with such intractable integrations of latent variables. By using VI, we finally obtain the lower bound of the CNCE loss in Eq. (5), which is tractable to compute.
>
> $m$ actually consists of latent tensor factors (as described in Line 132) and the size of $m$ is $D R$, where $D$ is the tensor order and $R$ is the tensor rank. The rank serves as a trade-off between model complexity and expressive power, which is usually treated as a hyper-parameter. In tensor decomposition, these tensor factors are crucial to learn latent correlations of the high-order data. Taking tensor completion applications as an example, once we learn these tensor factors, we are able to predict missing entries given very sparsely observed data.
>
> ## W2: Contribution.
>
> Our main contribution is to establish a new tensor decomposition model. As **Reviewer buha** suggested, there is no previous literature trying to use EBMs to model tensor data. Our model shows the potential to deal with diverse types of applications in tensor decompositions. Although both tensor decompositions and EBMs are existing models, it is generally hard to learn EBMs for such high dimensional data in the presence of latent factors. To enable efficient learning processes, we establish the variant of CNCE loss, as well as ad hoc network architectures, that finally result in a powerful TD model.
>
> ## Q1: Why not directly model $\phi(x; \theta)$ and $p_\theta(x)$.
>
> Please see our response to *W1*.

---

> > ### Author Response · Authors · 2023-08-16
> >
> > Dear Reviewer 32MU,
> >
> > Could we kindly know if the responses have addressed your concerns and if further explanations or clarifications are needed? Your time and efforts in evaluating our work is much appreciated.

---

> > > ### Comment · Reviewer_32MU · 2023-08-21
> > >
> > > The authors' rebuttal appropriately addresses my concern. I believe this work would contribute to the community in the future. In light of this, I would raise my score.

---

> > > > ### Author Response · Authors · 2023-08-21
> > > >
> > > > Thanks very much for your feedback! We will improve the work following your comments.

---

### Official Review · Reviewer_buha · 2023-07-05

**Soundness:** 3 good
**Presentation:** 3 good
**Contribution:** 3 good
**Rating:** 7
**Confidence:** 3

**Summary:**

The paper proposes an innovative approach for non-linear tensor decomposition. It utilizes the deep energy-based model (EBM) to model the joint energy function of tensor observations and latent factors. This design enables a more flexible decomposition without the need for explicitly defining the interaction between the latent factor and the commonly used Gaussian prior. Furthermore, the expressive nature of the energy function allows for the modeling of side information, such as timestamps.

**Strengths:**

The paper presents a clear motivation and introduces innovative and sophisticated solutions. To my knowledge, this is the first work to adopt Evidence-Based Medicine (EBM) for tensor decomposition without the need for an explicit design of the decomposition form.

Furthermore, the adaptability of the proposed method for dynamic tensors is impressive. The experimental setup is robust and comprehensive, demonstrating the thoroughness of the research.

**Weaknesses:**

1. Instead of using NON-FAT or BCTT for experiments on dynamic tensors' decomposition and visualization, it would be more appropriate to choose THIS-ODE[1] as the baseline. While NON-FAT and BCTT are designed for modeling temporal dynamics of factors in a tensor, THIS-ODE aligns with the proposed model by having an entry-wise dynamic design.

2. It appears that the objective function involves entry-wise importance sampling, and completing the posterior will require another round of entry-wise sampling. I am unsure if this will lead to training and inference costs becoming problematic, especially in scenarios with a large number of entries. It would be beneficial to have a more in-depth discussion about the scalability aspect.


[1] Decomposing Temporal High-Order Interactions via Latent ODEs, Shibo Li, Robert M. Kirby, and Shandian Zhe, The 39th International Conference on Machine Learning, 2022

**Questions:**

See weakness

**Limitations:**

See weakness

---

> ### Author Rebuttal · Authors · 2023-08-09
>
> ## W1: Comparison with THIS-ODE.
>
> Thanks to the reviewer for pointing out this reference, which is very relevant to our model. We would like to add some discussions about it in the manuscript and compare it with our model in experiments. However, its official implementation runs very slow and we are not able to complete very sufficient experiments due to the time limit. Here is a brief comparison about the computational time of several models, running on a single RTX A5000 GPU. Apart from THIS-ODE, we also list the computational time of CTGP, NNDTN and NONFAT, which perform better than other baselines. We test on the *Air* dataset, setting batch size to 128 and tensor rank to 5, and report the running time of one epoch by averaging on 10 runs. For our model, we set $\nu = 20$. For NONFAT and THIS-ODE, the default settings are used.
>
> |                         | CTGP            | NNDTN           | NONFAT          | THIS-ODE        | EnergyTD        |
> | ----------------------- | --------------- | --------------- | --------------- | --------------- | --------------- |
> | Time/Epoch (in seconds) | 1.17 $\pm$ 0.30 | 2.18 $\pm$ 0.04 | 2.51 $\pm$ 0.13 | 464. $\pm$ 131. | 5.30 $\pm$ 0.37 |
>
> As we can see, our model is slower than NONFAT, but much faster than THIS-ODE. While NONFAT is faster for each epoch, it converges much slower than our model. As suggested by the paper of NONFAT, we run 10,000 epochs in order to get a good performance. For our model, only 400 epochs is sufficient to converge.
>
> For THIS-ODE, it takes more than 30 hours to run 100 epochs on the *Air* dataset, and more than 40 hours to run 50 epochs on the *Click* dataset. Due to the time limit, we do not run more epochs. Moreover, the learning process shows that the model almost converges at this point, so the results are also convincing. The preliminary results are listed below and we will definitely conduct more sufficient experiments later. Our model outperforms THIS-ODE with much lower computational cost.
>
>
> |          | RMSE              |                   |                    |                   | MAE               |                   |                   |                   |
> | -------- | ----------------- | ----------------- | ------------------ | ----------------- | ----------------- | ----------------- | ----------------- | ----------------- |
> | *Air*    | Rank 3            | Rank 5            | Rank 8             | Rank 10           | Rank 3            | Rank 5            | Rank 8            | Rank 10           |
> | THIS-ODE | 0.588 $\pm$ 0.003 | 0.612 $\pm$ 0.000 | 0.586 $\pm$ 0.000  | 0.578 $\pm$ 0.000 | 0.434 $\pm$ 0.002 | 0.451 $\pm$ 0.000 | 0.426 $\pm$ 0.000 | 0.424 $\pm$ 0.000 |
> | EnergyTD | 0.302 $\pm$ 0.008 | 0.291 $\pm$ 0.006 | 0.300 $\pm$ 0.012  | 0.283 $\pm$ 0.004 | 0.184 $\pm$ 0.006 | 0.177 $\pm$ 0.003 | 0.172 $\pm$ 0.006 | 0.184 $\pm$ 0.003 |
> | *Click*  |                   |                   |                    |                   |                   |                   |                   |                   |
> | THIS-ODE | 1.408 $\pm$ 0.009 | 1.409 $\pm$ 0.008 | 1.405 $\pm$ 0.005. | 1.405 $\pm$ 0.008 | 0.846 $\pm$ 0.009 | 0.835 $\pm$ 0.007 | 0.846 $\pm$ 0.002 | 0.843 $\pm$ 0.007 |
> | EnergyTD | 1.396 $\pm$ 0.003 | 1.385 $\pm$ 0.003 | 1.356 $\pm$ 0.001  | 1.357 $\pm$ 0.001 | 0.777 $\pm$ 0.003 | 0.775 $\pm$ 0.003 | 0.772 $\pm$ 0.002 | 0.773 $\pm$ 0.001 |
>
> Currently, we have not got very faithful trajectory estimates using the simulation data with THIS-ODE
>
> ## W2: Training and inference cost.
>
> Firstly, for the training, the computational cost mainly comes from the evaluation of the energy function. In particular, we need to evaluate the energy function for $\nu$ times, where $\nu$ is the number of conditional noises and is typically 20 in our experiments. The time complexity is $\mathcal{O}(\nu B ( D R H + L H^2))$, where $B$ is the batch size, $\nu$ is the number of conditional noises, $H$ is the number of hidden units per layer, $L$ is the number of layers and $D$ is the tensor order. Since we are using very small NNs, e.g., several MLP layers with hundreds of hidden units each, the computational cost is not large. From the table above, we can see that our model is slower than NONFAT, but much faster than THIS-ODE. More importantly, since the forward passes for each particle can be computed in parallel, we believe the computational speed can be easily improved using parallel programming libraries such as `JAX` and `functorch`.
>
> Then, for the inference, apart from traditional MCMC samplers, we can also perform a very effective grid search to find MAP estimates in parallel. This can be much faster than the THIS-ODE model, which uses time-consuming diffusion processes for training and inference.

---

> > ### Comment · Reviewer_buha · 2023-08-11
> >
> > Thanks for the detailed response! I still strongly support this work.

---

> > > ### Author Response · Authors · 2023-08-11
> > >
> > > Thanks for your positive feedbacks! We will improve the manuscript following the suggestions.

---

### Author Rebuttal · Authors · 2023-08-09

We thank all the reviewers for their constructive comments and suggestions. Below we respond to them respectively.

---

> ### Comment · Area_Chair_s3C7 · 2023-08-18
> **Questions**
>
> Thank you for your responses.
>
> Below are not criticism of the quality of the paper. It would be helpful to understand the positioning of the paper a bit better.
>
> 1. Why is the proposed method a tensor decomposition method? I understand that it gives a  tensor completion. However, since it is a nonlinear neural network model, the prediction is not multi linear in the factors $m$ and the dimension of the factors $R$ is not related to the rank of the underlying tensor. Please clarify this. To me it seems more natural to think about the factors $m$ as embedding vectors of categorical inputs.
>
> 2. Why model the joint distribution p(x, m) instead of conditional p(x|m)? I understand that by treating the factors $m$ as random variables, we get the uncertainty estimate. However, parameter uncertainty doesn’t seem to be the main highlight of the paper. It is not clear to me if the complexity of marginalizing over $m$ is worth the trouble.
>
> Best,
> AC

---

> > ### Author Response · Authors · 2023-08-20
> >
> > Dear AC,
> >
> > Thanks very much for your fruitful comments and questions.
> >
> > **Q. Why is the proposed method a tensor decomposition method?**
> >
> > We agree with your assessment about the connection between our model and latent embeddings. Traditional tensor decompositions are also special cases of such embeddings with multi-linear mappings. We use the term **Tensor Decomposition** since our model also factorizes the tensor into factors for each mode, e.g., $z^1, z^2, z^3$. Besides, the non-linear mappings adopted in this work include traditional multi-linear mappings as special cases. For example, for an order-3 rank-1 tensor with $\boldsymbol{m} = [z^{1}_1, z^{2}_1, z^{3}_1]$, if we use a single MLP layer with polynomial activation $(\boldsymbol{w}^\intercal \boldsymbol{m})^3$, it already contains the multi-linear term $w_1 w_2 w_3 z^{1}_1 z^{2}_1 z^{3}_1$. Indeed, this extension of multi-linear mappings to non-linear mappings has been studied by several works in the tensor decomposition field, including Gaussian processes [1] and neural networks [2,3], as we discussed in the related work.
> >
> > Moreover, we think the latent factor size $R$ in our work shares some similarities with traditional tensor rank. Firstly, the size of the latent embedding $\boldsymbol{m}$, namely $DR$, is determined by $R$ and represents the intrinsic dimension of the data. Secondly, the energy function can be decomposed into summation of *rank-1* components as in CP or Tucker. Consider a rank-$R$ case, where $\boldsymbol{m} = [\boldsymbol{m_1}, \dots, \boldsymbol{m_R}]$ with
> > $\boldsymbol{m_i} = [z^1_i, z^2_i, z^3_i], i = 1, \dots, R$. For one layer MLP, the energy function is $f(x, m) = x \cdot \sum_{i=1}^R \boldsymbol{w_i}^\intercal \boldsymbol{m_i}$, where $\boldsymbol{w}_i$ are rows of MLP layer weights. In this case, each $\boldsymbol{w}^\intercal_i \boldsymbol{m}_i$ can be regarded as a rank-1 component. For deeper MLP layers, the relations are similar. However, we admit $R$ is different from traditional tensor ranks. In fact, in [1,3], $R$ is termed *number of factors*, while in [2], $R$ is termed *rank*. We will add more clarifications about this issue in the manuscript.
> >
> > - [1]. Zhe, el al. Distributed flexible nonlinear tensor factorization. NeurIPS’16.
> > - [2]. Liu el al. Costco: A neural tensor completion model for sparse tensors. KDD’19.
> > - [3]. Fang, et al. Streaming Bayesian deep tensor factorization. ICML’22.
> >
> > **Q. Why model the joint distribution $p(x, m)$ instead of conditional $p(x \mid m)$?**
> >
> > Generally, the computational cost of our model is $\nu$ times larger than traditional ones. However, since the network architecture is relatively small and these computations for $\nu$ particles are parallelizable, it is not a big cost using modern hardwares. Even without parallel computation, the computing time of our model is acceptable, as shown in our response to **Reviewer buha and Duah**. Moreover, since the marginalization is tackled through the variational approach (e.g, Eq. (5)), the additional computational cost is mainly from the usage of intractable EBM. Using the EBM brings great flexibility to our model and yields superior empirical results.
> >
> > We model the joint distribution since such undirected mappings are generally more succinct in capturing probabilistic dependencies than direct ones. As we discussed between Line 39-49 in the manuscript, modeling the conditional $p(x \mid m)$ requires additional assumptions about the likelihood and the prior, which could be unavailable in real applications. When dealing with $p(x, m)$, we add minimum assumptions by modeling the Gibbs distributions jointly. To the best of our knowledge, it is the first attempt to jointly model such multi-way data probabilities. Empirically, our model outperforms CoSTCo [3] and NNDTN, which are two baselines using such conditional $p(x \mid m)$, even CoSTCo employs specially designed CNNs and our model only uses MLPs.
> >
> > If there are any unclear issues in our response or if you have any further questions, we are more than happy to answer them.

---

### Decision · Program_Chairs · 2023-09-21

**Decision:**

Accept (poster)

**Comment:**

Looks like a good paper. Accept.